# Introducing Well-Defined Nanowrinkles in CVD Grown Graphene

**DOI:** 10.3390/nano9030353

**Published:** 2019-03-04

**Authors:** Tim Verhagen, Barbara Pacakova, Martin Kalbac, Jana Vejpravova

**Affiliations:** 1Department of Condensed Matter Physics, Faculty of Mathematics and Physics, Charles University, Ke Karlovu 5, 121 16 Prague 2, Czech Republic; 2J. Heyrovsky Institute of Physical Chemistry of the CAS, v.v.i., Dolejskova 2155/3, CZ-182 23 Prague 8, Czech Republic; barbara.pacakova@ntnu.no (B.P.); martin.kalbac@jh-inst.cas.cz (M.K.); 3Department of Physics, Norwegian University of Science and Technology (NTNU), Høgskoleringen 5, NO-7491 Trondheim, Norway

**Keywords:** CVD graphene, transfer, ruga, wrinkle, ripple, Raman spectroscopy, AFM

## Abstract

The control of graphene’s topography at the nanoscale level opens up the possibility to greatly improve the surface functionalization, change the doping level or create nanoscale reservoirs. However, the ability to control the modification of the topography of graphene on a wafer scale is still rather challenging. Here we present an approach to create well-defined nanowrinkles on a wafer scale using nitrocellulose as the polymer to transfer chemical vapor deposition grown graphene from the copper foil to a substrate. During the transfer process, the complex tertiary nitrocellulose structure is imprinted into the graphene area layer. When the graphene layer is put onto a substrate this will result in a well-defined nanowrinkle pattern, which can be subsequently further processed. Using atomic force and Raman microscopy, we characterized the generated nanowrinkles in graphene.

## 1. Introduction

The transfer of either exfoliated or chemical vapor deposition (CVD) grown graphene to another substrate is one of the key steps during the fabrication of graphene-based devices. The creation of rugae [1], a single state of a corrugated material configuration in which graphene can be mostly observed as exhibiting either wrinkles, ripples, folds, wrinklons or crumples, during the transfer is generally seen as unwanted, as these topographic features deteriorate the properties of graphene [2,3]. Several techniques were developed to improve the transfer of graphene as perfect as possible. Especially, the introduction of an all-dry transfer (whether or not using h-BN and annealing) resulted in devices with a minimum of these unwanted features [4,5].

However, those “unwanted topographic features” can be very suitable to further explore graphene’s potential. One of the most significant changes due to the presence of these rugae is a change in the pyramidalization angle, which describes the angle between the σ- and π-orbitals of graphene, and which is a measure for the chemical reactivity of carbon allotropes [6,7,8]. As the pyramidalization angle is highly dependent on the local curvature, a larger reactivity can be expected around the rugae. The chemical reactivity of the rugae can be further controlled via the doping level [9]. As they are locally delaminated from the substrate, the influence of the substrate doping is reduced. Furthermore, rugae can act as a generator for pseudo magnetic fields [10] or for anisotropic transport properties [11].

Besides changing these fundamental properties of graphene, a homogeneous, local deformation of graphene can act as a trap for liquids and gases, as graphene is an impenetrable membrane for most liquids and gases [12]. Besides acting as a storage space, these nanoreservoirs could also be an ideal chemical reactor for two dimensional (2D) chemistry [13] or used as a model system to study 2D confined liquids or the effect of a 2D liquid–membrane interface [14].

As shown recently by Hallam et al. [15], the strain in the graphene layer which is induced during the after-growth cooling is released during the removal of the Cu foil during the Cu etching step. Depending on the stiffness of the polymer, the strain within the graphene layer can be released via rugae. For CVD transferred graphene, these rugae most often consist of networks of wrinkles which have a width ranging between one and hundreds of nm, a height below 15 nm, a length (much) larger than 100 nm and an aspect ratio larger than 10. Although it is straightforward to induce some of these topographic corrugations during the transfer of graphene, it is a challenge to create these topographic features homogeneously in the whole transferred graphene layer.

The actual type of corrugation that is created during the transfer can be predicted if the properties of the used polymer for transfer are known. Using rugae mechanics, it can be shown that the shear modulus of both graphene and the polymer, the adhesion energy, the mismatch strain between the polymer and graphene, the orientation of graphene’s grain and defects edges and the structure of the used polymer are the dominant factors that determine the types of rugae that can be observed when the graphene is transferred to its final substrate [16,17]. For many polymers these parameters are known, and this information can be used to fine-tune the final rugae landscape.

In this article we show that by transferring graphene using nitrocellulose as a transfer polymer, we can induce a homogeneous landscape of nanoscale wrinkles in the whole graphene sheet, in which topography is induced from the structure of the used nitrocellulose polymer. This easy and straightforward method dramatically improves the production of large homogeneous areas of graphene with nanoscale wrinkles, which are created in a controllable and reproducible way. The ability to create large areas of graphene with the desired topographical properties improves the study of the influence of rugae on the electrical, optical and chemical properties of graphene, as there is no need to extensively search for large, homogeneous areas with the desired type of rugae.

## 2. Materials and Methods

Graphene samples were grown by the CVD method as reported previously [18]. In brief, the polycrystalline copper foil was heated to 1000 °C and annealed for 20 min under a flow of 50 standard cubic centimeters per minute (sccm) H_2_. The copper foil was exposed to 30 sccm CH_4_ and 50 sccm H_2_ for 20 min, after which the copper foil was cooled to room temperature.

On top of the foil, a nitrocellulose layer (NC) (Collodion solution for microscopy, 2% in amyl acetate, Sigma Aldrich 09817, St. Louis, Missouri, MO, USA), was spincoated with 2700 revolutions per minute (rpm) for 30 s and air dried, which results in a thickness of the NC of approximately 50 nm. The transfer of the 1 × 1 cm^2^ graphene/NC layer is the same as is used for the transfer with PMMA. In short; the Cu foil was etched away using FeCl_3_ (copper etchant type CE-100, Transene Company Inc., Danvers, MA, USA), and the graphene/NC sample was fished using a Si/SiO_2_ wafer from the solution and subsequently washed several times using highly purified water. Finally, the graphene/NC layer was fished using the final substrate, which was carefully cleaned using acetone and isopropanol, and dried with nitrogen gas. Residual NC can be easily removed from the graphene by washing the sample several times with methanol. The samples were dried with nitrogen gas and no further annealing steps were performed.

Prepared samples were characterized by atomic force microscopy (AFM) and Raman spectroscopy. The Raman spectra were obtained at a WITec alpha300R spectrometer (WITec Wissenschaftliche Instrumente und Technologie GmbH, Ulm, Germany) equipped with a piezo stage. Raman spectral maps were measured with 2.33-eV (532-nm) laser excitation, a laser power of approximately 1 mW, a grating of 600 lines/mm and lateral steps of 1 μm in both directions. The laser was focused on the sample with a 100× objective. The D, G and 2D modes of the graphene monolayer were analyzed using pseudo-Voigt peak profiles, where one, three (G_1_, G_2_ and D’) and two (2D_1_ and 2D_2_) peak profiles were used to fit the D, G and 2D modes respectively. The G_1_, G_2_, 2D_1_ and 2D_2_ modes were used to account for the delaminated graphene, which was created by the NC transfer [19].

AFM was measured using PeakForce quantitative nano-mechanical mapping (PeakForceQNM) and imaging in tapping mode, using a Dimension Icon microscope (Bruker Inc., Billerica, MA, USA). Images were captured using Bruker Scanasyst-Air probes (k = 0.4 N/m, f_0_ = 70 kHz, nominal tip radius = 2 nm) at room temperature in air. High quality images were processed in the standard way using Gwyddion software [20], applying line-by-line first order leveling and scar removal.

To determine the overall orientation of wrinkles, Gwyddion’s 2D FFT analysis (Gwyddion, Czech Metrological Institute, Brno, Czech Republic) is used. Grain analysis was used to evaluate the surface coverage by the winkles (*A_wr_*), and to determine their geometry and width (*w_wr_*). Surface characterization was used for the estimation of the roughness (*R_a_*), root-mean-square roughness (*R_q_*), and surface area (SA). The profile height distribution was analyzed to determine the median peak-to-valley distance (*R_tm_*), of the wrinkle graphene layer topography, which was used as the wrinkle height. The topography AFM image was cut into individual lines using home-built procedures, in order to identify all the minima and maxima attributed to the highest parts of the wrinkles and the lowest parts in the inter-wrinkle valleys. Then, the median difference of all maxima and minima was calculated as *R_tm_* = *p_μ_* − *v_μ_*, where *p_μ_* is the median wrinkle height and *v_μ_* is the median valley depth, calculated over all the lines in the image.

## 3. Results

Figure 1a shows an AFM image of the CVD grown graphene transferred with NC on Si/SiO_2_ substrate. It can be clearly seen that the graphene is not flat, but a clear, fine landscape of graphene wrinkles is present. Graphene is wrinkled very homogeneously, as shown from the 2D FFT in Figure 1c, within the whole graphene layer and consisted everywhere of a fine structure composed of connected wrinkles.

Figure 1e shows an AFM image of a NC layer directly spincoated on a Si/SiO_2_ substrate without graphene. On the 2 × 2 um^2^ scale, the NC layer was relatively smooth. In Figure 1f a close up of this layer is shown where the very fine, complex tertiary structure of the NC is visible with a height of less than 1 nm, which resembles the landscape of graphene wrinkles presented in Figure 1a.

Homogeneously distributed wrinkles were found to be uniformly covering almost 65% of graphene sheet area (see Table 1). The large-scale wrinkles and folds, which are often present when graphene is transferred with PMMA [11], were rarely observed. If there were large folds, they were wrinkled in the same way as the whole graphene layer. On the large scale, there was no orientation preference, indicating an isotropic compression of the graphene layer, and wrinkles were oriented equally in all directions, as can be seen from the 2D FFT image (Figure 1c). If there was some preferential orientation direction of the wrinkles, the 2D FFT image would not be spherical, but asymmetric in the direction of preferential orientation. Furthermore, the intensity of the 2D FFT sphere was homogeneous and no bright rings in the sphere are visible. This indicates that the distance between two wrinkles varied from wrinkle to wrinkle and no repeating wrinkle pattern was present. The wrinkle height, which is represented by the *R_tm_* value, reached 2.5 nm and wrinkle width varies from 6 to 15 nm.

### Raman Spectroscopy

Figure 2 shows a typical Raman spectrum of a CVD graphene monolayer transferred with NC. While the G and 2D modes of graphene are clearly visible, almost no D mode is visible indicating that no significant amount of defects were created in graphene during the NC transfer. Furthermore, the G and 2D modes are rather asymmetric, and therefore fitted with a G_2_ and 2D_2_ sub-band. The presence of these sub-bands confirms the results of the AFM analysis, indicating that a significant part of the graphene was delaminated [19]. Finally, the intensity of the G and 2D modes is approximately the same, which was caused by the relatively high doping level of the graphene, as can be seen in Figure 3e [21].

The Raman data were further analyzed in order to get deeper insight in the charge and strain management in the graphene monolayer with regularly distributed nanowrinkles. Raman spectral maps of the Raman shift of the G_1_ and 2D_1_ modes, and the full width at half maximum (FWHM) of the 2D_1_ mode are shown in Figure 3a–c. The Raman shift of the G_1_ and 2D_1_ modes was not homogeneous, and well-defined regions with a smaller or larger Raman shift are visible. These regions were caused by the different Cu grains of the Cu foil, onto which the graphene was grown. As shown before [21], the actual strain and doping varies depending on the crystal orientation of the Cu grain in the Cu foil. This variation in strain and doping was visible as a variation in the actual Raman shift and FWHM of the G and 2D modes. Using well known relations describing the relation between strain, doping and the measured Raman shifts of the G and 2D modes, an actual estimation of the strain and doping can be made [22,23]. As a first estimate, a biaxial strain distribution was assumed. As shown in Figure 3d, also in the calculated strain map, clear regions with a different strain were be observed, whereas in the doping map shown in Figure 3e these regions showed a much more homogeneous doping.

The FWHM of the 2D mode, which acts as a probe of the nanometer-scale flatness [24] is shown in Figure 3c. The FWHM of the 2D_1_ mode of the NC transferred graphene was rather homogeneous and there was no obvious dependence visible, as could be seen for the Raman shift of the G_1_ and 2D_1_ modes. This is an indication that during the NC transfer, the wrinkles as shown in the AFM images in Figure 1 were homogeneously induced in the graphene layer. Figure 3f shows the distribution of the FWHM of the 2D_1_ mode, which shows a narrow distribution of FWHM values around 48 cm^−1^, where extremely flat graphene can have a FWHM of the 2D mode well below 20 cm^−1^ [5].

However, the correlation between the calculated strain and FWHM of the 2D_1_, as shown in Figure 3h, shows that there might be a correlation between the strain and the FWHM of the 2D_1_ mode. For increasing compressive strain, the FWHM of the 2D mode slightly decreased, whereas for increasing tensile strain, the FWHM of the 2D mode has a larger scatter and was not increasing.

This trend is also visible in the correlation plot between the Raman shift of the G_1_ – 2D_1_ (colored points) and G_2_ – 2D_2_ (black points) modes, as shown in Figure 3g, where the FWHM of the 2D_1_ mode is indicated via a color code. The attached graphene to the substrate (colored points) was relatively doped and a large strain variation was present. Interestingly, these points were clustered close to the blue line indicating graphene with a strain value close to zero, but with an increasing doping level, had a lower FWHM of the 2D_1_ mode than points that had some tensile strain.

As the FWHM of the 2D_1_ mode of the NC transferred graphene seemed to be strain dependent, this suggests that there is also the uniaxial strain component present within the graphene layer [25]. Although all the wrinkles are oriented randomly with respect to each other, it is not surprising that graphene feels locally around a single wrinkle this uniaxial strain component due to the large amount of one dimensional wrinkles. Subsequently, Raman spectroscopy will probe the contribution of all the different local strain components [24]. No significant influence of the doping level on the FWHM of the 2D_1_ mode is expected, as the FWHM did not change significantly when the doping level was changed.

The delaminated graphene (black points) had a very low doping level and showed the same large strain variation as observed for the attached graphene.

## 4. Discussion

The rugae landscape in CVD grown graphene that is transferred to another substrate can be controlled by the used polymer. Up to now, the choice of polymer was often influenced by the possibility to completely remove the polymer after the transfer, so that as clean as possible graphene could be obtained. Here, we show that NC transferred graphene, besides utilizing a very clean transfer method [15], also results in the creation of a very fine pattern of wrinkles in the whole transferred graphene sample, which is evidenced by AFM and an extreme broadening of the 2D mode in Raman spectra.

The relatively rigid NC preserves the strain in graphene that is built up during the CVD growth as can be seen in the maps of the Raman shift of the G and 2D modes (Figure 3). Areas with a well-defined Raman shift are clearly recognizable. The variation of the Raman shift in the different areas is mostly caused by the variation in strain, as can be seen in the Figure 3, which strongly depends on the Cu grain orientations in the Cu foil where graphene was grown on.

The structure of NC shows at the nanoscale a complex network of nanoscale wrinkles, as is visible in Figure 1f. When the Cu foil is etched away below the graphene/NC stack, the rigid NC polymer layer is able to preserve the compressive, growth-induced strain as was already observed by Hallam et al. [15]. However on the nanoscale, some strain is released locally in the graphene layer and thereby graphene locally copies the complex tertiary NC structure and forms a complex nanowrinkle network, as shown in Figure 1a.

The observed wrinkles in the AFM images are not caused by polymer residuals. The strong asymmetric G and 2D modes indicates that a significant part of the graphene layer is delaminated [15]. Furthermore, the FWHM of the 2D mode acts as a probe of the nanometer scale flatness [24]. The extremely broad FWHM of the 2D mode of the NC transferred graphene films indicates that graphene is not flat on the nanometer scale, and the wrinkles observed with AFM are thus caused solely by the graphene layer.

## 5. Conclusions

We have shown that when NC is used as the polymer for the transfer of CVD grown graphene from the Cu foil to another substrate, randomly distributed regular nanowrinkles are imprinted in the graphene layer. Our approach opens up many new possibilities to control and exploit the properties of graphene with nanometer spatial resolution, like the doping level or chemical functionalization. Moreover, it allows to discover the potential of nanoscale reservoirs on wafer scale.

## Figures and Tables

**Figure 1 nanomaterials-09-00353-f001:**
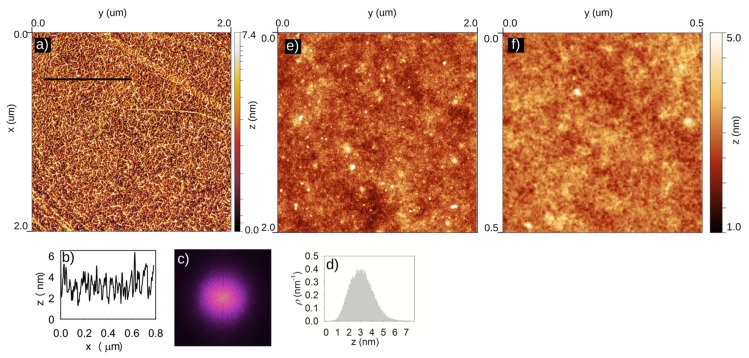
The atomic force microscopy (AFM) topography image of the wrinkled NC transferred graphene (**a**) with the cross-section (black line) in panel (**b**). The 2D FFT image of the AFM topography image (**c**), showing no orientation preference of wrinkles, and the height histogram of the topography image (**d**). (**e**) Large-scale AFM image of NC on a Si/SiO_2_ substrate. (**f**) A close up AFM image of the NC on a Si/SiO_2_ substrate as shown in (**e**).

**Figure 2 nanomaterials-09-00353-f002:**
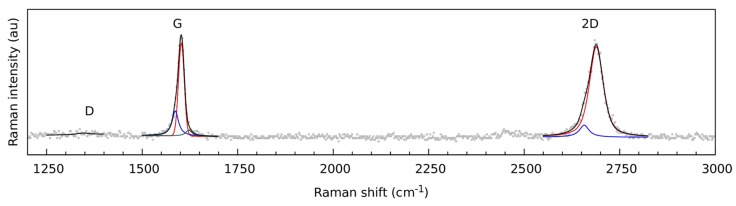
Typical Raman spectrum of graphene, which is transferred using NC. The experimental data are marked by gray dots and fits of the individual bands are shown in red for the G_1_ and 2D_1_ components, blue for the G_2_ and 2D_2_ components and grey for the D’ component, respectively. The resulting curve (sum of the individual components) is represented by a solid black line.

**Figure 3 nanomaterials-09-00353-f003:**
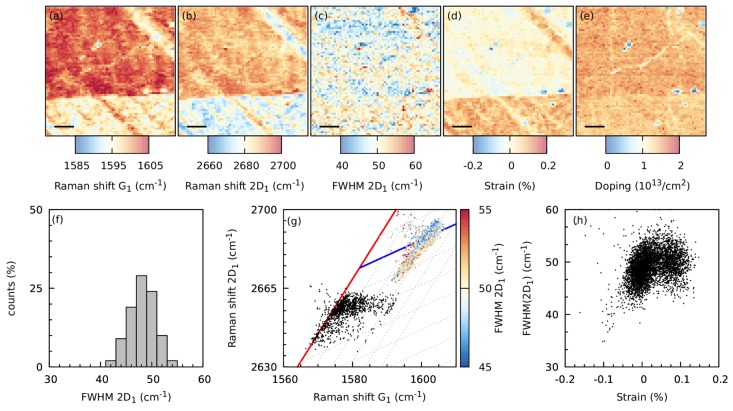
Raman spectral maps of the fitted Raman shift of the G_1_ (**a**) and 2D_1_ (**b**) modes, the full width at half maximum (FWHM) of the 2D_1_ mode (**c**) and the calculated strain (**d**) and doping (**e**) for graphene transferred with NC. (**f**) A histogram of the fitted FWHM of the 2D_1_ mode. (**g**) Correlation plot of the Raman shift of the 2D_1_ and 2D_2_ modes as a function of the Raman shifts of the G_1_ and G_2_ modes, respectively. The (G_2_ – 2D_2_) data points are shown in black, whereas the (G_1_ – 2D_1_) data points are colored, which represents the FWHM of the 2D_1_ mode. (**h**) The correlation between the FWHM of the 2D_1_ mode versus the calculated, biaxial strain.

**Table 1 nanomaterials-09-00353-t001:** The basic characteristics of the NC transferred graphene, determined from the AFM images. Roughness (*R_a_*), root-mean-square roughness, (*R_q_*), wrinkle height (*R_tm_*), surface area (SA), area covered with the wrinkles (*A_wr_*) and width, (*w_wr_*).

*R_a_* (nm)	*R_q_* (nm)	*R_tm_* (nm)	SA (%)	*A_wr_* (%)	*w_wr_* (nm)
0.83	1.03	2.5	105	65	6–15

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
