# Peer review of "Introducing Well-Defined Nanowrinkles in CVD Grown Graphene"

_nanomaterials, 2019, doi:10.3390/nano9030353_

Round 1
Reviewer 1 Report
The control over the properties of the graphene is highly desirable from the point of view of applications. It is especially important to control the properties on the scale larger than purely local one, obtaining significant areas of graphene modified in homogeneous manner. The present manuscript presents a method of transferring the CVD graphene from the Cu foil to the Si/SiO2 substrate using nitrocellulose. The crucial feature of this method is that it leads to particular uniformly wrinkled topography of graphene. The process itself is described first. The obtained graphene on final substrate is than characterised using two techniques. The first one is AFM technique applied to visualise the wrinkling on the surface (the image of graphene is compared with the image of the nitrocellulose on the same substrate) The second technique is Raman scattering spectroscopy used to estimate the strain and doping level. The study is careful and the presentation of the data and discussion is clear and convincing. The importance of the results is well explained and the subject is certainly worthy of investigation.
I recommend the manuscript for publication in Nanomaterials. Below I list some minor remarks which the Authors might take optionally into account before:
* It might be useful to mention what was the temperature at which AFM study was performed (was it a room-temperature study in air)?
* Since it appears that the average wrinkle length l_wr was not estimated (as it can be deduced from N.A.), maybe this quantity might be removed from Table 1 (or some explanation might be presented why it was not possible to determine this quantity).
* Maybe some histogram of wrinkle widths and of wrinkle-wrinkle distances could also be interesting (if it can be constructed out of the data without excessive difficulties).
Author Response
We are very thankful to the reviewers for their comments, which helped us to improve our manuscript.
Review 1
The control over the properties of the graphene is highly desirable from the point of view of applications. It is especially important to control the properties on the scale larger than purely local one, obtaining significant areas of graphene modified in homogeneous manner. The present manuscript presents a method of transferring the CVD graphene from the Cu foil to the Si/SiO2 substrate using nitrocellulose. The crucial feature of this method is that it leads to particular uniformly wrinkled topography of graphene. The process itself is described first. The obtained graphene on final substrate is than characterised using two techniques. The first one is AFM technique applied to visualise the wrinkling on the surface (the image of graphene is compared with the image of the nitrocellulose on the same substrate) The second technique is Raman scattering spectroscopy used to estimate the strain and doping level. The study is careful and the presentation of the data and discussion is clear and convincing. The importance of the results is well explained and the subject is certainly worthy of investigation.
I recommend the manuscript for publication in Nanomaterials. Below I list some minor remarks which the Authors might take optionally into account before:
* It might be useful to mention what was the temperature at which AFM study was performed (was it a room-temperature study in air)?
We have added the conditions of the afm measurements (room temperature, air).
* Since it appears that the average wrinkle length l_wr was not estimated (as it can be deduced from N.A.), maybe this quantity might be removed from Table 1 (or some explanation might be presented why it was not possible to determine this quantity).
We have removed the average wrinkle length from the table, as it is quite difficult to obtain this number, see also next point.
* Maybe some histogram of wrinkle widths and of wrinkle-wrinkle distances could also be interesting (if it can be constructed out of the data without excessive difficulties).
It is quite difficult to extract these parameters from afm images, as there are many wrinkles randomly oriented, very close to each other.
Reviewer 2 Report
see attached file

Author Response
We are very thankful to the reviewers for their comments, which helped us to improve our manuscript.
Review 2
The manuscript “Introducing Well-defined Nanowrinkles in CVD Grown Grown Graphene” by Verhagen et. al. describes a study of the morphology and quality of CVD graphene transferred on substrates using nitrocellulose as the supporting layer. The developed method affords clean graphene films with controlled wrinkled morphology. The experiments are well planned and performed, the materials are thoroughly characterized using atomic force microscopy and Raman. The manuscript is clearly written and I recommend it for publication after minor revision:
1. Please, indicate the typical size of transferred graphene in this study.
We transfer typically ~1x1 cm2 graphene.
2. Consider presenting the images in Figure 1a and 1e in the same scale, i.e. 2 um as in Fig. 1a. This will make the comparison of the images easier.
We thank the referee for pointing out this inconsistency and we have updated the size of fig. 1e
3. Line 178 – delete the entire line or edit the sentence on lines 178-180 as it does not read well.
We modified the sentence for better clarity.
4. Suggestions for references to be added:
- reviews on transfer of CVD graphene:
Zhao, G. et. al., The Physics and Chemistry of Graphene-on-surfaces Chem. Soc. Rev. 2017, 46, 4417-4449.
Chen, M.; Haddon, R. C.; Yan, R.; Bekyarova, E., Advances in Transferring Chemical Vapor Deposition Graphene: a Review. Mater. Horiz. 2017, 4, 1054-1063.
- in addition to ref. [4] add the references:
Haddon, R. C., pi-Electrons in Three-Dimensions. Acc. Chem. Res. 1988, 21, 243-249.
Niyogi, S.; Hamon, M. A.; Hu, H.; Zhao, B.; Bhowmik, P.; Sen, R.; Itkis, M. E.; Haddon, R. C., Chemistry of Single-Walled Carbon Nanotubes. Acc. Chem. Res. 2002, 35, 1105-1113.
Prof. Haddon did pioneering work on pyramidalization angle and its effect on the chemical reactivity of conjugated carbon materials, which deserves to be acknowledged.
We thank the referee for pointing to this references. We have added them to the manuscript
Reviewer 3 Report
The manuscript entitled "Introducing Well-defined Nanowrinkles in CVD Grown Graphene" by Verhagen et al. reports creation of nanowrinkles using nitrocellulose to transfer chemical vapor deposition (CVD) grown graphene to a Si/SiO2 substrate. The authors fabricated CVD graphene on copper foil and transferred it onto a Si/SiO2 substrate with nitrocellulose. They characterized wrinkles in the transferred graphene by using atomic force microscopy and Raman scattering spectroscopy measurements. The paper proposes a simple and effective way to create homogeneously distributed nanowrinkles in graphene, and might give high impact in the field of nanomaterials science especially graphene science. Thus, I think the paper deserves publication in Nanomaterials. However, I recommend the revision concerning the following point for improvement of the paper.
1) The authors claim at lines 189-190 that there is a uniaxial strain component present within the graphene layer due to the large amount of one dimensional wrinkles. However, they also claim at lines 127-128 that the graphene is wrinkled with no orientation preference, indicating an isotropic compression of the graphene layer. These descriptions are incompatible with each other. So, it needs to clearly explain the relation between the descriptions.
Minor remarks:
- line 179: Figure 4 (h) à Figure 3 (h)?
- caption of Figure 3 (a-e): The explanation of scale bars is needed.
Author Response
We are very thankful to the reviewers for their comments, which helped us to improve our manuscript.
Review 3
1) The authors claim at lines 189-190 that there is a uniaxial strain component present within the graphene layer due to the large amount of one dimensional wrinkles. However, they also claim at lines 127-128 that the graphene is wrinkled with no orientation preference, indicating an isotropic compression of the graphene layer. These descriptions are incompatible with each other. So, it needs to clearly explain the relation between the descriptions.
We did not specify clearly on which scales we were applying the claims. On the AFM picture in Fig. 1, we claim that on a 2x2 um2 (and larger) scale, the distribution of the wrinkles is random. But locally, around one wrinkle, the strain is not randomly distributed but has a preferred strain direction, influence mostly by the one-dimensional wrinkle. This local, preferred strain direction is subsequently probed by the Raman measurements, where the laser beam will probe all these local contributions
We have updated the text in the manuscript to make this seemingly discrepancy more clear.
Minor remarks:
- line 179: Figure 4 (h) à Figure 3 (h)?
- caption of Figure 3 (a-e): The explanation of scale bars is needed.
We have updated both points.